# Open Hardware Implementation of Real-Time Phase and Amplitude Estimation for Neurophysiologic Signals

**DOI:** 10.3390/bioengineering10121350

**Published:** 2023-11-23

**Authors:** José Ángel Ochoa, Irene Gonzalez-Burgos, María Jesús Nicolás, Miguel Valencia

**Affiliations:** 1Biomedical Engineering Program, Physiological Monitoring and Control Laboratory, CIMA, Universidad de Navarra, Avda Pio XII 55, 31080 Pamplona, Spain; jochoamarti@unav.es (J.Á.O.); igonzalezb@unav.es (I.G.-B.); mnicolas@unav.es (M.J.N.); 2IdiSNA, Navarra Institute for Health Research, C/Irunlarrea, 31008 Pamplona, Spain; 3Institute of Data Science and Artificial Intelligence, Universidad de Navarra, Campus Universitario, 31009 Pamplona, Spain

**Keywords:** open hardware, real-time, phase and amplitude estimation, physiologic signals, adaptive stimulation, real-world implementation

## Abstract

Adaptive deep brain stimulation (aDBS) is a promising concept in the field of DBS that consists of delivering electrical stimulation in response to specific events. Dynamic adaptivity arises when stimulation targets dynamically changing states, which often calls for a reliable and fast causal estimation of the phase and amplitude of the signals. Here, we present an open-hardware implementation that exploits the concepts of resonators and Hilbert filters embedded in an open-hardware platform. To emulate real-world scenarios, we built a hardware setup that included a system to replay and process different types of physiological signals and test the accuracy of the instantaneous phase and amplitude estimates. The results show that the system can provide a precise and reliable estimation of the phase even in the challenging scenario of dealing with high-frequency oscillations (~250 Hz) in real-time. The framework might be adopted in neuromodulation studies to quickly test biomarkers in clinical and preclinical settings, supporting the advancement of aDBS.

## 1. Introduction

The emergence of closed-loop feedback systems in medicine gives rise to responsive, dynamic, and adaptive therapies with the ability to smartly self-regulate, thus breaking with the traditionally used open-loop therapies and a large part of their limitations [1,2,3,4]. In the field of neurology, the concept of brain stimulation involves employing diverse modalities (such as cortical stimulation, deep brain stimulation, and responsive neurostimulation) through electrodes placed on the scalp or implanted in deep brain structures to treat neurological disorders. Currently, brain stimulation is predominantly used for the treatment of movement disorders like Parkinson’s disease (PD). In this context, previous studies have already proposed a closed-loop approach for the improvement of deep brain stimulation (DBS) therapy, and the results are auspicious [5,6,7,8,9,10]. However, its scope is expanding to include the treatment of neuropsychiatric conditions such as depression and schizophrenia. In the case of epilepsy, closed-loop brain stimulation emerges as an alternative for patients who are unsuited to the conventional surgical resection of the epileptogenic zone [11,12,13].

Despite the current lack of comprehensive understanding of optimizing detection and stimulation parameters for personalized benefits, the treatment is gaining importance in clinical practice due to its proven clinical efficacy and potential to provide an individualized and dynamically adapted therapy based on a person’s intracerebral electrophysiology [13,14].

Dynamic adaptivity arises when stimulation targets dynamically changing states, such as the oscillatory phase or amplitude of certain frequencies [4]. In this case, it is necessary to deliver stimulation in close temporal proximity to the event of interest. This calls for a reliable and fast causal estimation of the phase and amplitude of the signals of interest. For this purpose, several strategies have been developed [8,10,15,16,17,18]. Nevertheless, most of them remain in the theoretical/computational domain [18,19,20,21], deal with frequencies below 20 Hz [10,22], or rely on the use of very sophisticated equipment [16,17]. All these limitations pose a threat to advancing and evaluating potential strategies for the development of new aDBS paradigms.

In this context, open hardware initiatives such as Arduino [23] could play a pivotal role in rapidly prototyping and developing preclinical setups designed to assess potential biomarkers and stimulation/recording paradigms. Its versatility, affordability, and user-friendly nature make it an accessible tool for scientists, researchers, and students interested in exploring various aspects of neuroscience. Arduino’s contributions to neuroscience span a range of applications, from building experimental setups to creating innovative solutions to study and understand the complexities of the nervous system [24,25,26,27].

Here, we present an open hardware, Arduino-based implementation for the phase and amplitude computation of neurophysiological signals in real-time. Our approach exploits the concepts of resonators and Hilbert filters embedded in an open-hardware electronic prototyping platform to obtain a real-time causal estimator of the phase and amplitude of neurophysiological signals [18,28,29,30,31].

To illustrate the performance and versatility of the proposed approach, we tested it on a physiologic-like synthetic signal, recordings of epileptic seizures from a preclinical mouse model, movement activity in relation to parkinsonian tremor, and local field potentials recorded from the subthalamic nucleus (STN) from PD patients. The results show that this system can track in real-time the phase and amplitude of neurophysiological signals even in the challenging scenario of dealing with the real-time processing of activities above 200 Hz.

## 2. Materials and Methods

The proposed implementation exploits the concepts of resonators and Hilbert filters embedded in an open-hardware prototyping platform to obtain a real-time, causal estimator of the phase and amplitude of a neurophysiological signal in a frequency range of interest. Filter design and deployment were accomplished using Matlab and Simulink (The MathWorks, Inc., Natick, MA, USA) on two different Arduino implementations.



*Filter design and Hardware implementation*



The instantaneous phase and amplitude of the neurophysiological signal in a frequency range of interest are based on obtaining a real-time estimate of the analytic signal (Figure 1a). To perform this, we first used a resonator filter tuned to the frequency of interest (IIR filter block in Figure 1a), followed by a Hilbert transform filter that shifts π⁄2 radians for each frequency component (Hilbert transform filter block in Figure 1a). By combining the output of these two processes, it is possible to obtain an estimation of the analytical signal from which the instantaneous phase and amplitude are calculated.

In other words, from a wide band real signal *x*(*t*), the resonator filter generates *x_f_*(*t*), a narrow band signal that contains the energy of *x*(*t*) around the frequency of interest *f*. The Hilbert transform filter induces a shift of π⁄2 radians for each frequency component of *x_f_*(*t*) and results in the time-resolved estimation of the Hilbert transform of *x_f_*(*t*), *x_HT_*(*t*). At this point, it is important to limit the bandwidth of the *x_f_*(*t*) (ideally to be mono-frequential) so the Hilbert transform is well-defined.

Combining *x_f_*(*t*) + *x_HT_*(*t*) results in the analytic version of *x*(*t*) as follows:$$\tilde{x}\left(t\right)=x_{f}\left(t\right)+{ix}_{HT}\left(t\right)=A\left(t\right)e^{jϴ\left(t\right)}$$
where *A*(*t*) and *ϴ*(*t*) represent the instantaneous amplitude and phase of *x*(*t*), respectively.

Both filters, the resonator and Hilbert transform were designed using *filterDesigner*, a graphical user interface (GUI) of the MATLAB Signal Processing Toolbox™. The filters are exported as transfer functions in the time domain, so the real-time implementation permits estimations sample-by-sample for the instantaneous amplitude and phase of the signal *x*(*t*). The blocks *amplitude computing* and *phase computing* in Figure 1a implement the computations of the instantaneous amplitude and phase, respectively, according to the following:$$A\left(t\right)=\sqrt{{x_{f}(t)}^{2}+{x_{HT}(t)}^{2}}$$
$$\theta \left(t\right)=arctan\left(\frac{x_{HT}(t)}{x_{f}(t)}\right)$$

Having instantaneous phase *ϴ*(*t*) and amplitude *A*(*t*) estimates, it is straightforward to establish certain conditions and thresholds for the said parameters and define the markers that could potentially trigger the stimulator.

Unlike conventional offline non-casual procedures, the proposed implementation takes data and provides an estimation for every sample that arrives in the pipeline. As a result, it is suitable for its deployment in hardware systems to be used for real-time closed-loop experiments.

Here, we used Simulink™ to exploit it into the following two Arduino implementations: the Arduino Nano IoT (NANO) and Arduino DUE (DUE) boards. These two boards include the analog-to-digital converter (ADC) ports needed to sample the signal and digital input/output ports (DIO) to deliver the marker, which is amenable to deploying our framework for real-time processing (main steps are shown in Figure 1b). The Arduino NANO board has a SAMD21 Cortex^®^-M0+ 32bit low power ARM MCU microcontroller with an 8/10/12 bit ADC input, which allows the signal to be sampled and to run the Simulink model at 1200 Hz. The DUE has an AT91SAM3 × 8E microcontroller with 12 analog inputs, 2 analog outputs, and 54 digital I/O pins. For our purposes here, we set the sampling frequency of 1 of the 12 analog inputs to 20,000 Hz and the system did not experience any delay in running the model in real-time (more information about the NANO and DUE implementations can be found at https://store-usa.arduino.cc/products/arduino-nano-33-iot and https://store-usa.arduino.cc/products/arduino-due, respectively, access date 21 November 2023).



*Benchmark setup*



To exemplify our approach, we implemented a benchmark setup that included a Power1401 625 kHz 16-bit data acquisition interface controlled by Spike2 10.06 software (Cambridge Electronic Design Ltd., Cambridge, UK) and coupled to an Arduino board that implemented the close-loop. The Power1401 is a high-performance data acquisition interface capable of recording waveform and digital (event) data and simultaneously generating waveform and digital outputs for real-time, multi-tasking experimental control.

Examples of physiologic signals were delivered through one of the Power1401 digital-to-analogic (DAC) ports and recorded by an analog-to-digital (ADC) Power1401 port while simultaneously routed to the Arduino (Figure 1c, blue arrows). The closed-loop system (i.e., Arduino board) then processed the signal and delivered TTL pulses into a digital input port of the Power1401 (Figure 1c, red arrow). By performing this, we obtained a synchronized recording that included the original signal together with the TTL pulses. Finally, the Spike2 files were converted and analyzed offline through custom-made routines implemented in Matlab.

The accuracy of the phase estimation was performed by computing the circular statistics of the phase of the physiological signal at the time points defined by the TTLs delivered using the Arduino system [32]. We estimated the length of the mean resultant vector R and the angular deviation. R is a crucial quantity for the measurement of circular spread and hypothesis testing in directional statistics, whereas the angular deviation was introduced as an analog to the linear standard deviation. The closer R was to 1, the more concentrated the data sample was around the mean direction. In addition, we computed the Rayleigh test, which asks how large the resultant vector length R must be to indicate a non-uniform distribution and is particularly suited to detecting a unimodal deviation from uniformity.

## 3. Results

The proposed framework and its implementation take into consideration the principles of usability and applicability in real-world scenarios. To illustrate this, we tested our approach on four types of signals:(1)A physiologic-like synthetic signal inspired by Rosenblum et al. 2021 [18].(2)The resting tremor velocity signal from the index finger in PD patients taken from PhysioNet.org [33,34].(3)Electrocorticographic recordings of epileptic seizures from a mouse model of epilepsy [35].(4)Local field potentials recorded from deep-brain stimulation (DBS) electrodes placed in the subthalamic nucleus of PD patients [36].

### 3.1. Synthetic Signal from Rosenblum et al. 2021 [18]

The synthetic signal proposed by [18] shows modulations in both frequency and amplitude as an attempt to resemble the behavior of physiological activities. For this signal, we first explored the effect of modifying the amplitude threshold (thr = 1.97, 2.23, 2.46) in the detection of a fixed value of the phase (−1.2 rad) (Figure 2a, left panel). As expected, different thresholds for the amplitude resulted in fewer TTLs delivered by Arduino while maintaining good stability in the detection of the target phase.

Then, we gradually increased the value of the main frequency of the oscillator (and modified the parameters of the resonator and Hilbert filters accordingly) to evaluate the performance of the two Arduino implementations. For 8 Hz, the phase distribution detected by the NANO results at an R-value of 0.98 (*p* < 0.001 Rayleigh test for circular statistics), illustrated a good performance for the stability of the detection of the instantaneous phase of the signal (blue histogram in the right panel). For frequencies 20, 40, and 80 Hz, the width of the distribution widened progressively and R-values decayed to 0.97, 0.96, and 0.92, respectively (orange, yellow, and purple histograms). Nevertheless, for 80 Hz, the power of DUE allowed R = 0.98 to be recovered (green histogram), and even in the demanding scenario of dealing with frequencies of 250 Hz the DUE showed a good performance (R = 0.94, light blue histogram). All these results demonstrate the good performance of the proposed implementation in processing the physiologic-like synthetic signal proposed by Rosenblum et al. (Figure 2b,c, Table 1).

Finally, we explored the stability of the detection as a function of the target phase in the (−π, π] range. By fixing the frequency of the oscillator but varying the target phase, we were able to assess the performance of detection and show that there were no significant differences (Figure 2d).

### 3.2. Rest Tremor Velocity

After testing our approach in a synthetic signal, we moved to a real-world scenario. We evaluated the performance of the system to detect the phase of the rest tremor velocity signal measured in the index finger of PD patients. Although the intrinsic dynamics and biological variability of the signal could represent a challenge for detection, the relatively low frequency of the tremor (~4 Hz) and oscillatory-like behavior of the signal allowed for good performance for the detection (Figure 3).

The database contains two different sets of recordings. The first set of recordings corresponds to patients with high amplitude tremor (HT), whereas the second set corresponds to patients with little or no tremor (low amplitude tremor, LT). For the whole set of HT patients (n = 8), the detection showed an R-value > 0.72, with a mean value of R = 0.9 and a standard deviation of 0.11. For the LT group (n = 7), performance dropped significantly because, in most cases, there was no tremor signal, and the implementation tried to “catch” the noise. Interestingly, and back to the HT group, only two out of eight patients showed R-values below 0.9. The reason for this behavior is that these two patients showed a multi-frequency tremor signal that resulted in an unstable detection of the phase of the signal that jumped from one mode of oscillation to the other.

### 3.3. Mouse Model of Epilepsy

Continuing with neurophysiological signals but now increasing the amount of noise and difficulties in terms of the irregularity of the waveforms, we explored the capabilities of the system to detect the phases of electrophysiological recordings during epileptic seizures in a mouse model of epilepsy [35]. Recordings consisted of 11 seizures with a pre-ictal period followed by the onset of ictal activity with the main frequency in the range of 4–8 Hz. Despite the asymmetry of the waveforms and the inherent noise of the recording, the system was capable of providing an accurate estimation of the phase for the oscillatory activity that characterizes the seizures (Figure 4e, Table 1). Although in most of the cases (7 out of 11 analyzed seizures), the performance was very good (R-value > 0.75), in some other cases (4 out of 11), this value dropped but remained above 0.55. This was due to the presence of noise and the sub-optimal (and intentional) selection of the amplitude threshold. This caused the Arduino NANO to detect spurious TTLs in the preictal period that could be easily removed and, thus, increase the R-value. Nevertheless, during the seizure, the system was capable of dealing with the quasi-triangular, asymmetric, and frequency-varying nature of the oscillation that characterized the seizure.

### 3.4. Local Field Potentials from the Subthalamic Nucleus of Parkinson’s Disease Patients

Finally, we aimed to process local field potentials recorded from the STN of PD patients undergoing DBS therapy. The oscillatory activity in the STN of patients with PD is a prominent feature of the disorder. In the absence of medication (OFF state), abnormal oscillations, typically in the beta frequency (around 13–30 Hz) and high-frequency (~250 Hz) ranges, are often observed and are correlated with motor symptoms. Although classical stimulation (open loop) can effectively alleviate and provide symptomatic relief in many patients, some studies suggest dynamically targeting these oscillations in a phase-specific manner could improve DBS performance.

As a result, we first evaluated the capability to target specific phases of the oscillatory activity in the beta range. The results showed that the proposed implementation is capable of dealing with the intrinsic noise of the STN recordings and providing robust estimates of the phase for the pathological beta activity (see Figure 5d,e) for a representative example.

For the HFO, the challenge is even higher, as computations must be completed in less than one cycle (i.e., less than 4 milliseconds). Despite this tremendous requirement, the results show that the Arduino NANO is capable of sampling the signal, estimating the phase/amplitude of the HFO, and delivering the corresponding TTLs with good performance (see Figure 5f,g for a representative example).

It is important to note that the signal delivered to the Arduino is the raw data and not filtered versions of the electrophysiological signal were around beta and HFO ranges; both Arduinos included the filtering, analytical signal estimation, and real-time phase and amplitude detection stages, thus increasing the translational potential of the proposed system. The results from the whole set of patients (corresponding to 8 recordings from [36]) reveal that the mean performance for the beta range was R = 0.78 (std = 0.07) and R = 0.79 (std = 0.05) for the HFOs (Figure 5h, Table 1).

**Table 1 bioengineering-10-01350-t001:** Summary of the results for the four different scenarios evaluated. R refers to the mean resultant vector length across all the recordings. The *p*-value corresponds to the Rayleigh test that assesses how large the resultant vector length R must be to indicate a non-uniform distribution and is particularly suited for detecting an unimodal deviation from uniformity. ang-dev refers to the mean angular deviation computed across the different recordings for each modality of the signal.

Data	Target	Arduino Board	R	*p*-Value	Ang-Dev (Rad)
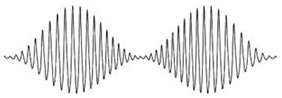 synthetic signal from [18]	8 Hz	NANO	0.977	<0.001	0.214
	20 Hz	NANO	0.972	<0.001	0.236
	40 Hz	NANO	0.956	<0.001	0.295
	80 Hz	NANO	0.919	<0.001	0.402
	80 Hz	DUE	0.979	<0.001	0.203
	250 Hz	DUE	0.945	<0.001	0.332
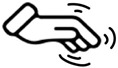 tremor	HT	NANO	0.904	<0.001	0.379
	LT	NANO	0.610	<0.001	0.872
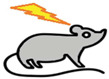 seizure	Ictal Activity	NANO	0.737	<0.001	0.710
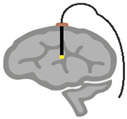 DBS	Beta	NANO	0.778	<0.001	0.657
	HFO	DUE	0.789	<0.001	0.646

## 4. Discussion

Here, we proposed and deployed a closed-loop framework that was capable of effectively estimating the instantaneous phase and amplitude of physiological signals in a wide range of frequencies. We demonstrated that our approach is suitable for its use in a variety of relevant scenarios by illustrating our approach in a synthetic signal modulated by amplitude and frequency; considering the tremor activity from PD patients; electrocorticographic data from seizures in a mouse model of epilepsy; and finally in processing the local field potentials recorded from the STN of PD patients.

In all four scenarios, the system can lock the TTL’s to the phase of the signal of interest independently of the type of signal or the frequency of interest in a frequency range that covers from a few Hertzs to frequencies above 200 Hz. In all cases, the R-value is above 0.7, with the only exception being the case where no signal is present or shows a very low amplitude (low tremor group in the tremor data set). On the one hand, the Rayleigh test shows a very significant deviation from a uniform distribution of the phases. On the other hand, the phase histograms, together with the low angle deviation values, reveal a well-defined preference for the target phase, thus demonstrating the good performance of the framework. It is noteworthy that the maximal mean angular deviation (excluding the low tremor group) is 0.7 rads, which roughly corresponds to 1/10th of the cycle, meaning that most of the TTLs fall in a narrow phase range around the target phase. In addition, we show that for a specific frequency of interest, the performance is stable across the whole set of phases. This allows the peak to be targeted alongside the though, the peak-to-trough transition, or the trough-to-peak transition with high specificity, thus permitting an exploration of the effect of the stimulation depending on the phase of stimulation. Finally, and contrary to most previous reports, we set up a real-world demonstrator providing proof of its capabilities and flexibility to deal with the intrinsic noise and instabilities of physiologic signals in a wide range of frequencies from low (~4 Hz) to very high frequencies (~250 Hz).

Some solutions for closed-loop stimulation, such as NeuroPace for epilepsy [37], AspireRS 106 for vagus nerve stimulation [38], or the investigational Activa PC+S, Summit RC+S [39], and Percept PC system for movement disorders, are capable of sensing and stimulating in close-loop. However, the wider adoption of these approaches is awaiting advances in chronic validation [40,41].

The accurate and timely detection of symptoms in brain disorders is critical to enable closed-loop neuromodulation, and it typically requires the use of suitable biomarkers [42,43]. By embracing Arduino, an open, broadly accepted, and accessible hardware implementation for rapid prototyping [23], we expect that our framework may be adopted in adaptive neuromodulation studies to quickly test biomarkers in clinical and preclinical settings, supporting the advancement of aDBS. Among them, the phase and/or amplitude closed-loop stimulation seems to be a good candidate [5,8,10,17,44]. Instead of relying on complicated, highly specialized, or proprietary hardware, here we describe a real-world, reconfigurable open hardware implementation for the real-time estimation of phase and amplitude for physiological signals suitable for incorporation into the toolbox of any clinical or preclinical neuromodulation study.

## 5. Conclusions

Here, we present an open hardware implementation for the phase and amplitude computation of neurophysiological signals in real-time. After testing our approach in a synthetic data set, we illustrated its performance by computing the phase and amplitude for several scenarios, such as epileptic seizures from a preclinical mouse model, resting tremors, and STN activity (including beta and HFO) from PD patients. The results show that the proposed approach can track in real-time the phase and amplitude of neurophysiological signals even in the challenging scenario of dealing with oscillations above 250 Hz.

By embracing a hardware implementation that is open, widely accepted, and easily accessible for rapid prototyping, we anticipate that this framework could be utilized in aDBS studies. This could facilitate the rapid testing of biomarkers in both clinical and preclinical environments, thereby contributing to the progress of aDBS.

## Figures and Tables

**Figure 1 bioengineering-10-01350-f001:**
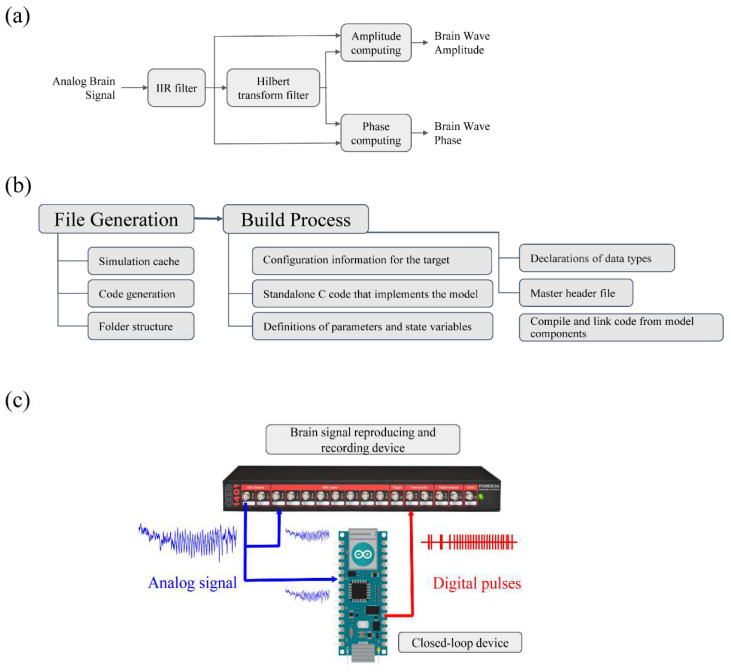
**System implementation.** (**a**) Block diagram for real-time phase and amplitude computing based on exploding the concepts of resonators and Hilbert filters to obtain an estimate of the analytic signal of the incoming physiological activity. (**b**) Simulink workflow for the deployment of the proposed workflow on the Arduino NANO and DUE platforms (**c**) Benchmark set-up to assess the performance of the proposed framework.

**Figure 2 bioengineering-10-01350-f002:**
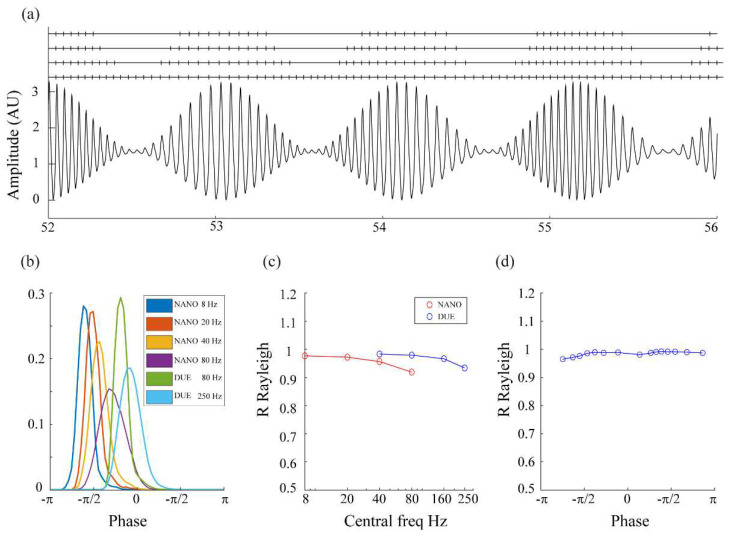
**Performance of the proposed framework on the physiologic-like synthetic signal modulated by frequency and amplitude.** (**a**) Examples of detection of the phase for the synthetic signal are presented in Rosenblum et al., 2021, for different amplitude thresholds. (**b**) Normalized histogram of the detected phases for increasing frequencies of the oscillator; while the NANO can deal with frequencies up to 80 Hz, the power of DUE allows it to reach frequencies in the range of 250 Hz (please note that the target phase for each frequency has been shifted so histograms do not overlap). (**c**) R-values correspond to the distribution of phases at different frequencies detected using Arduino NANO (8, 20, 40, 80 Hz) and DUE (40, 80, 160, 250 Hz). (**d**) R-values associated with the detection of different phases (data for an Arduino NANO implementation on a synthetic signal with a main frequency of 20 Hz).

**Figure 3 bioengineering-10-01350-f003:**
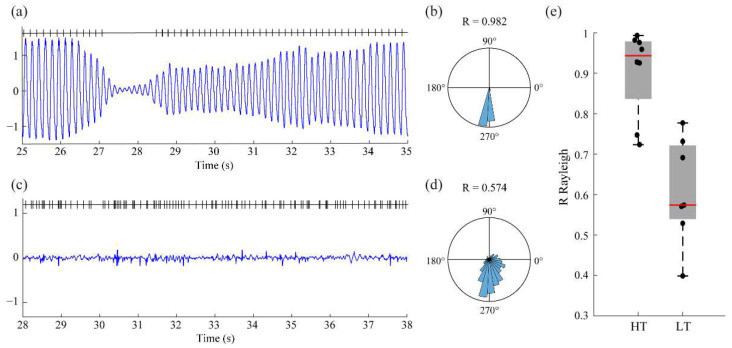
**Performance of the proposed framework on the real-time processing of tremor-related signals.** (**a**) Example and detection of the phase in the tremor signal of a representative subject from the HT group. (**b**) Phase plot and R-value corresponding to the panel (**a**). (**c**) Example and detection of the phase in the tremor signal of a representative subject from the LT group. (**d**) Phase plot and R-value corresponding to the panel (**c**). (**e**) Box plot representation of the R-values corresponding to the two sets of patients. Note that the performance of the framework in the LT group is lower than that of the RT group.

**Figure 4 bioengineering-10-01350-f004:**
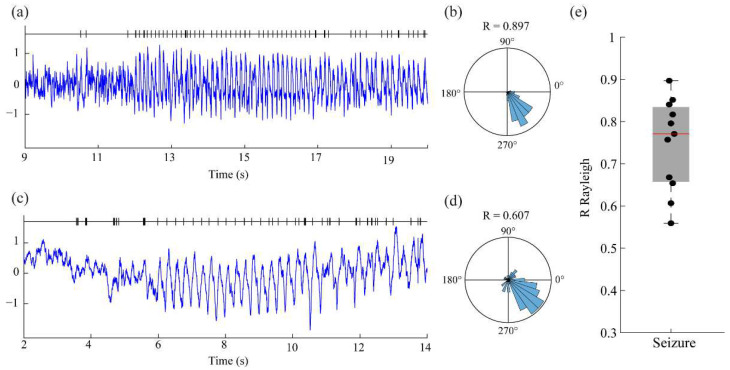
Performance of the proposed framework on the real-time processing of sharp waves that occur during a heat-induced seizure in a mouse model of epilepsy. (**a**–**c**) Raw data example of two different seizures. (**b**–**d**) Phase plot and R-value corresponding the to signal in (**a**,**c**), respectively. (**e**) Box plot representation of the R-values corresponding to the phase detection for the 11 seizures analyzed.

**Figure 5 bioengineering-10-01350-f005:**
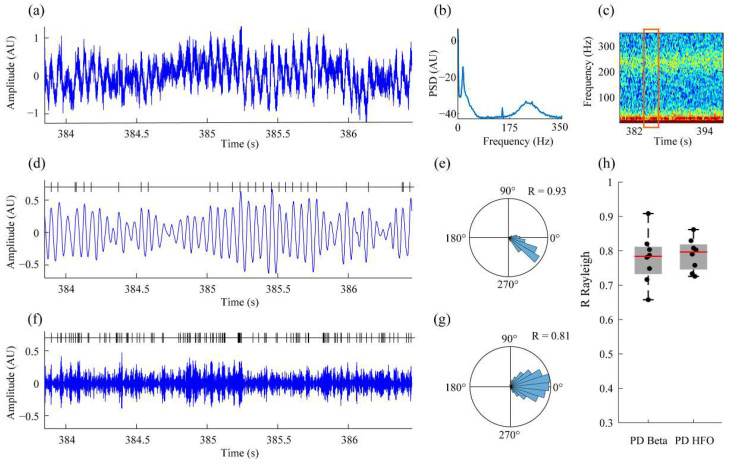
Performance of the proposed framework on the analysis of subthalamic local field potentials recorded from DBS electrodes in Parkinson’s disease patients. (**a**) Raw data example of one representative recording (OFF medication state). (**b**) Power spectral estimate of the STN activity showing the presence of two main oscillatory rhythms, one in the beta range (~20 Hz) and another in the HFO range (~230 Hz). (**c**) Spectrogram of the STN signal showing the time-frequency distribution of the beta and HFO activities. (**d**) Filtered version (~17.5 Hz, beta range) of the fragment in (**a**) together with the TTLs delivered by the Arduino NANO. (**e**) Polar plot of the phases computed from the TTL time stamps processed in the beta range. (**f**) Filtered version (~235 Hz, HFO) of the fragment in (**a**) together with the TTLs delivered by the Arduino DUE. (**g**) Polar plot of the phase distribution for the detection of HFOs. (**h**) Box plot representation of the R-values corresponding to the detection of the phase for the beta and HFO activities in human STN recordings.

## Data Availability

The data used here combine synthetic [18] and real neurophysiologic signals from previously published works [33,34,35,36]. The code for simulating the synthetic data in [18] is available from the corresponding author upon reasonable request. Resting tremor data in [33,34] are freely available at https://www.physionet.org/content/tremordb/1.0.0/. Electrophysiological recordings from PD patients in [36] are not publicly available due to privacy and ethical restrictions and could be available on request from the corresponding author. The recordings from the animal model of epilepsy in [35] are available from the corresponding author upon reasonable request.

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
