# Peer review of "Open Hardware Implementation of Real-Time Phase and Amplitude Estimation for Neurophysiologic Signals"

_bioengineering, 2023, doi:10.3390/bioengineering10121350_

Round 1
Reviewer 1 Report
Comments and Suggestions for Authors
The article describes the Real-Time Phase and Amplitude Estimation for Neurophysiologic Signals. The topic is interesting, although there are some issues that need to be addressed:
1) The introduction is poor and have to be extended about literature review.
2) Authors should provide more details about hardware and benchmark procedure
3) Discussed about measurement errors should be add
4) Can the authors confirm the validity of the results obtained by comparing them with other typical studies?.
5) Does authors tried applied FFT procedure to the obtained data? Are results are interesting ?
6) Fig. 2b . The data are overlapped and unreadable.
Author Response
Thank you very much for taking the time to review this manuscript.
Please find in the attached file the detailed responses and the corresponding revisions/corrections highlighted in the re-submitted manuscript file.

Reviewer 2 Report
Comments and Suggestions for Authors
This paper presents an open-hardware implementation that exploits the concepts of resonators and Hilbert filters embedded in an open-hardware platform. Application results show that the system can provide reliable estimation of the phase even in the challenging scenario of dealing with high-frequency oscillations (~250 Hz) in real-time.
The method in Section 2 is not described clearly. Some more explanations of Figure 1 should be given.
Author Response

(The authors gave the same response as above.)

Reviewer 3 Report
Comments and Suggestions for Authors
The authors have investigated different types of neurophysiological signals in real-time using an open hardware implementation .
The English sentences need to be revised and past tense:
English corrections:
To emulate real-world scenarios, we built a hardware setup that includes a system to replay and process different types….
…We have built
After testing our approach in a synthetic data set, ??
to obtain an estimate of the analytical signal …change in estimation
we implemented a benchmark setup that in….we have implemented
Among them, phase and/or amplitude locked closed-loop stimulation of physiologic signals seems to be a good candidate
The obtained results appear not clear . Please to include a diagram or scheme for better comprehension of results
The introduction doesn’t include the use and description of Arduino .Please cite this article :
KCIUK, Marek, et al. Intelligent Medical Velostat Pressure Sensor Mat Based on Artificial Neural Network and Arduino Embedded System. Applied System Innovation, 2023, 6.5: 84.
Comments on the Quality of English LanguageThe English sentences need to be revised and past tense:
English corrections:
To emulate real-world scenarios, we built a hardware setup that includes a system to replay and process different types….
…We have built
After testing our approach in a synthetic data set, ??
to obtain an estimate of the analytical signal …change in estimation
we implemented a benchmark setup that in….we have implemented
Among them, phase and/or amplitude locked closed-loop stimulation of physiologic signals seems to be a good candidate
T
Author Response

(The authors gave the same response as above.)

Round 2
Reviewer 2 Report
Comments and Suggestions for Authors
The authors have adequately addressed my comments and the revised manuscript can be accepted.